# Quantitative evaluation of *Mycobacterium abscessus* clinical isolate virulence using a silkworm infection model

Yasuhiko Matsumoto[1]*, Hanako Fukano[2], Naoki Hasegawa[3], Yoshihiko Hoshino[2]*, Takashi Sugita[1]

**1** Department of Microbiology, Meiji Pharmaceutical University, Tokyo, Japan, **2** Department of Mycobacteriology, Leprosy Research Center, National Institute of Infectious Diseases, Tokyo, Japan, **3** Department of Infectious Diseases, Keio University School of Medicine, Tokyo, Japan

* ymatsumoto@my-pharm.ac.jp (YM); yhoshino@niid.go.jp (YH)

## Abstract

*Mycobacterium abscessus* causes chronic skin infections, lung diseases, and systemic or disseminated infections. Here we investigated whether the virulence of *M. abscessus* clinical isolates could be evaluated by calculating the median lethal dose ($LD_{50}$) in a silkworm infection model. *M. abscessus* subsp. *abscessus* cells were injected into the silkworm hemolymph. When reared at 37˚C, the silkworms died within 2 days post-infection with *M. abscessus* subsp. *abscessus*. Viable cell numbers of *M. abscessus* increased in the hemolymph of silkworms injected with *M. abscessus*. Silkworms were not killed by injections with heat-killed *M. abscessus* cells. The administration of clarithromycin, an antibacterial drug used to treat the infection in humans, prolonged the survival time of silkworms injected with *M. abscessus*. The $LD_{50}$ values of 7 clinical isolates in the silkworm infection model were differed by up to 9-fold. The Mb-17 isolate, which was identified as a virulent strain in the silkworm infection model, induced more detachment of human THP-1-derived macrophages during infection than the Mb-10 isolate. These findings suggest that the silkworm *M. abscessus* infection model can be used to quantitatively evaluate the virulence of *M. abscessus* clinical isolates in a short time period.

## Introduction

The *Mycobacterium abscessus* complex (MABC) is a group of rapidly growing non-tuberculous mycobacteria that includes 3 subspecies: *M. abscessus* subsp. *abscessus*, *M. abscessus* subsp. *massiliense*, and *M. abscessus* subsp. *bolletii* [1–4]. The MABC causes chronic skin infections, lung diseases, and systemic or disseminated infections in immunocompromised patients [5–8]. The virulence of MABC has evolved through a stepwise adaptation to the host and soil environments [5, 9], and could thus vary between clinical isolates. Several mouse MABC infection models have been used to evaluate the efficacies of antibacterial drugs [10, 11]. Existing mouse models of MABC infection require several weeks to complete a single study, which is not convenient, especially for MABC virulence screening purposes. Thus, developing a model that permits a more rapid evaluation of MABC virulence is highly desirable.

**Funding:** This study was supported in part by grants from the Japan Agency for Medical Research and Development/Japan International Cooperation Agency (AMED) to YH (JP20fk0108064, JP20fk0108075, JP21fk0108093, JP21fk0108129, JP21fk0108608, JP21jm0510004, JP21wm0125007, JP21wm0225004, JP21wm0325003, JP22gm1610003, JP22wm0225022, JP22wm0325054), to HF (JP22wm0325054), and Y.M. (JP22wm0325054); and for Scientific Research (C) to YM (JP20K07022) from the Japan Society for the Promotion of Science (JSPS). The funders had no role in the study design, data collection, data analysis, decision to publish, or preparation of the manuscript.

**Competing interests:** The authors declare no conflict of interest.

Silkworms are invertebrate animals beneficial for use in experiments to reveal host-pathogen interactions [12–14]. A large number of silkworms can be reared in a small space compared with mammalian animals [15]. The 3Rs, replacement, refinement, and reduction, are important animal welfare principles for experiments using mammals [16]. Experiments using invertebrates are consistent with the concept of replacement. Because the silkworm is an invertebrate, fewer ethical issues are associated with the use of a large number of silkworms for experimentation compared with mammals. By exploiting this advantage of silkworms for infectious disease research, the median lethal dose ($LD_{50}$), which is the dose of a pathogen required to kill half of the animals in a group, can be determined to quantitatively compare the virulence of different strains [17, 18]. Silkworm infection models are used as initial screening systems to identify virulence-related genes in pathogenic microorganisms [19–22]. Silkworm infection models are therefore useful for comparing the virulence of microorganisms [15]. A silkworm infection model was established to evaluate anti-mycobacterial compounds using a type strain [23]. Virulence among *M. abscessus* clinical isolates based on the $LD_{50}$ values, however, has not been evaluated in a silkworm *M. abscessus* infection model.

In the present study, we compared the virulence of *M. abscessus* subsp. *abscessus* clinical isolates by calculating the $LD_{50}$ values in a silkworm infection model. Among the 7 clinical isolates evaluated, the extent of the virulence varied up to 9-fold. Furthermore, using the silkworm infection model, the *M. abscessus* subsp. *abscessus* Mb-17 isolate was identified as a highly virulent strain that exhibits higher cytotoxic activity against human THP-1 macrophages compared with the Mb-10 isolate. These findings suggest that the silkworm infection model is a rapid evaluation system for quantitatively estimating the virulence of *M. abscessus* subsp. *abscessus* clinical isolates.

## Materials and methods

### Reagents

Clarithromycin (Tokyo Chemical Industry Co., Ltd., Tokyo, Japan) was suspended in 0.9% NaCl solution (saline). Middlebrook 7H9 broth, Middlebrook 7H10 agar, and Middlebrook OADC enrichment were purchased from Becton, Dickinson, and Company (Sparks, MD, USA). Middlebrook 7H9 broth and Middlebrook 7H10 agar were supplemented with 10% Middlebrook OADC Enrichment.

### Bacterial strains and growth

The *M. abscessus* subsp. *abscessus* ATCC19977 strain and 7 clinical isolates (Mb-7, Mb-10, Mb-14, Mb-16, Mb-17, Mb-18, and Mb-22) were used in this study. The clinical isolates were obtained from sputum samples of patients infected with *M. abscessus* subsp. *abscessus* at the Keio University School of Medicine. This study was approved by the medical research ethics committee of the National Institute of Infectious Diseases (#1046) and by the Keio University School of Medicine Ethics Committee (#2008-0131-9 sai). Bacterial species were identified with a DDH Mycobacteria Kit (Kyokuto Pharmaceutical Industrial Co., Tokyo, Japan) [24] and multiplex polymerase chain reaction [25]. The *M. abscessus* subsp. *abscessus* strains were grown on a Middlebrook 7H10 agar plate at 37˚C. A single colony was then inoculated into 5 ml of Middlebrook 7H9 broth and incubated at 37˚C for 3 days.

### Infection experiments using silkworms

The silkworm infection experiments were performed as previously described [26]. Fifth instar larvae were reared on an artificial diet (Silkmate 2S, Ehime-Sanshu Co., Ltd., Ehime, Japan)

for 24 h. *M. abscessus* subsp. *abscessus* cells grown in Middlebrook 7H9 broth were collected by centrifugation and suspended in sterile saline. A 50-μl of sample solutions was administered to the silkworm hemolymph by injecting the silkworm dorsally using a 1-ml tuberculin syringe (Terumo Medical Corporation, Tokyo, Japan). Silkworms were injected with the *M. abscessus* subsp. *abscessus* cells ($1.4 \times 10^7$ cells per larva), and were incubated at 27˚C or 37˚C, and their survival was monitored.

The therapeutic activity of clarithromycin in silkworms was evaluated according to a previous study with slight modifications [26]. Either 50 μl of saline or 50 μl of an *M. abscessus* subsp. *abscessus* suspension was injected into the silkworm hemolymph. Clarithromycin (25 μg/g larva) was immediately injected into the silkworms. The silkworms were incubated at 37˚C, and their survival was monitored.

## Viable cell counts

Silkworms were injected with an *M. abscessus* subsp. *abscessus* cell suspension ($7 \times 10^6$ cells in 50 μl) and incubated at 37˚C. Hemolymph was harvested from the silkworm larvae through a cut on the first proleg at either 3 or 18 h post-infection [17]. The hemolymph was added to saline, and the solution was spread on a Middlebrook 7H10 agar plate. The agar plate was incubated at 37˚C for 3 days, and the colonies on the agar plate were counted.

## LD$_{50}$ measurement

The LD$_{50}$ values were determined according to a previous study, with slight modifications [27, 28]. *M. abscessus* subsp. *abscessus* cells grown in Middlebrook 7H9 broth were suspended in saline. Either a 2- or 4-fold dilution series of the bacterial suspension was prepared. The bacterial suspension ($4 \times 10^5$–$1 \times 10^8$ cells/50 μl) was injected into the silkworm hemolymph, and the silkworms were incubated at 37˚C. The number of surviving silkworms was counted at 48 h after infection. The LD$_{50}$ values were determined from the data of 3 experiments using a simple logistic regression model in Prism 9 (GraphPad Software, LLC, San Diego, CA, USA, https://www.graphpad.com/scientific-software/prism/).

## Cytotoxicity test using human THP-1–derived macrophages

To evaluate the cytotoxicities of *M. abscessus* subsp. *abscessus* clinical isolates against human THP-1-derived macrophages, a high-content imaging analysis was performed to count the number of cells attached to the polystyrene surface of a 96-well plate. Cell detachment from the well correlated with cell death [29]. Therefore, the cytotoxicities of the *M. abscessus* subsp. *abscessus* clinical isolates against THP-1–derived macrophages were determined by monitoring the cell detachment. Human THP-1 monocytes were cultured in RPMI1640 medium (Wako Pure Chemical Corporation, Osaka, Japan) supplemented with 10% fetal bovine serum (Thermo Fisher Scientific Inc., Waltham, MA, USA) and ampicillin (50 μg/mL) at 37˚C and 5% $CO_2$. Cells ($2 \times 10^4$ cells/well) were seeded into 96-well plates (Cell Carrier-96well Ultra microplate; PerkinElmer Inc., Waltham, MA, USA) and differentiated to macrophages with phorbol 12-myristate 13-acetate (10 ng/mL) for 72 h. The single monolayer THP-1 macrophages were infected by *M. abscessus* subsp. *abscessus* Mb-10 or Mb-17 isolates at a multiplicity of infection of 50, centrifuged at 1000 rpm for 5 min, and incubated for 4 h. To remove the extracellular bacteria, wells were washed twice with phosphate-buffered saline (PBS) and RPMI 1640 with amikacin (200 μg/mL) was added. After 2-h incubation, wells were washed twice with PBS and RPMI 1640 was added. The plates were incubated for 48 h at 37˚C and 5% $CO_2$. After incubation, infected cells were fixed with 4%(v/v) paraformaldehyde for 10 min at room temperature and washed 3 times with ice-cold PBS before treating them with 0.1%

Triton X-100 for 10 min, and then washing 3 times with PBS. The cells were stained with Hoechst 33258 (1:1000; Dojindo Molecular Technologies, Inc., Kumamoto, Japan) and HCS Cell Mask Deep Red (1:20000; Thermo Fisher Scientific Inc., MA, USA) for 25 min at room temperature and then washed 3 times with PBS. Stained cell images were obtained using a High-Content Imaging System Operetta CLS (PerkinElmer Inc.) with 40x Air/0.6 NA. The number of attached cells (cytoplasm with nuclei) was automatically calculated by Harmony software (PerkinElmer Inc.). Data are expressed as the mean ± the standard deviation (SD). Statistical analysis was performed with GraphPad Prism 9 (GraphPad Software).

## Statistical analysis

The statistical significance of differences between viable cell counts of *M. abscessus* subsp. *abscessus* in silkworm groups were determined by the Student *t*-test. Statistically significant differences between survival curves in the silkworm infection experiments were evaluated using a log-rank test in GraphPad Prism 9 (GraphPad Software). The statistical significance between Mb-10 and MB-17 isolates in the infection experiment using human THP-1 macrophages was calculated using the Turkey test with one-way ANOVA. Values of $P < 0.05$ were considered significant.

## Results

### Experimental conditions for the evaluation of *M. abscessus* subsp. *abscessus* virulence in silkworms

We first determined the experimental conditions for evaluating the virulence of *M. abscessus* subsp. *abscessus* in silkworms. In silkworm infection experiments, the rearing temperature is critical because it affects bacterial virulence and silkworm health [15]. The standard silkworm rearing temperature is 27˚C, and 37˚C corresponds to human body temperature. We previously reported that administration of *Staphylococcus aureus* (1 x $10^7$ cells) to silkworms killed them in 3 days under rearing conditions of 27˚C [19]. Within 48 h of injection of *M. abscessus* subsp. *abscessus* ATCC19977 strain (1.4 x$10^7$ cells), silkworms died at 37˚C, but did not die at 27˚C (Fig 1A and 1B). The viable cell count of *M. abscessus* subsp. *abscessus* in the silkworm hemolymph at 18 h post-infection at 37˚C was higher than that at 27˚C (Fig 1C). The $LD_{50}$ for *M. abscessus* subsp. *abscessus* ATCC19977 was 1.1 x $10^7$ cells under a 37˚C rearing condition (Fig 2). These results suggest that a 37˚C rearing condition is necessary for *M. abscessus* subsp. *abscessus*-induced silkworm death within 2 days.

### Virulence effect of *M. abscessus* subsp. *abscessus* proliferation in silkworms

Because silkworm death caused by *Porphyromonas gingivalis* in a previous report did not require proliferation, we assumed that the silkworm death was caused by shock not induced by the bacterial infection [30]. We tested whether *M. abscessus* subsp. *abscessus* proliferates in the silkworm body, and if proliferation is necessary for virulence. The *M. abscessus* subsp. *abscessus* viable cell count increased in the silkworm hemolymph at 18 h post-infection (Fig 3). The injection of autoclaved *M. abscessus* subsp. *abscessus* cells did not kill silkworms (Fig 4A). Clarithromycin, a bacteriostatic antibiotic, is used to treat clinical *M. abscessus* subsp. *abscessus* infections in humans [8, 31]. The administration of clarithromycin to silkworms infected with *M. abscessus* subsp. *abscessus* prolonged their survival time (Fig 4B). These results suggest that the virulence of *M. abscessus* subsp. *abscessus* against silkworms requires the growth of *M. abscessus* subsp. *abscessus*.

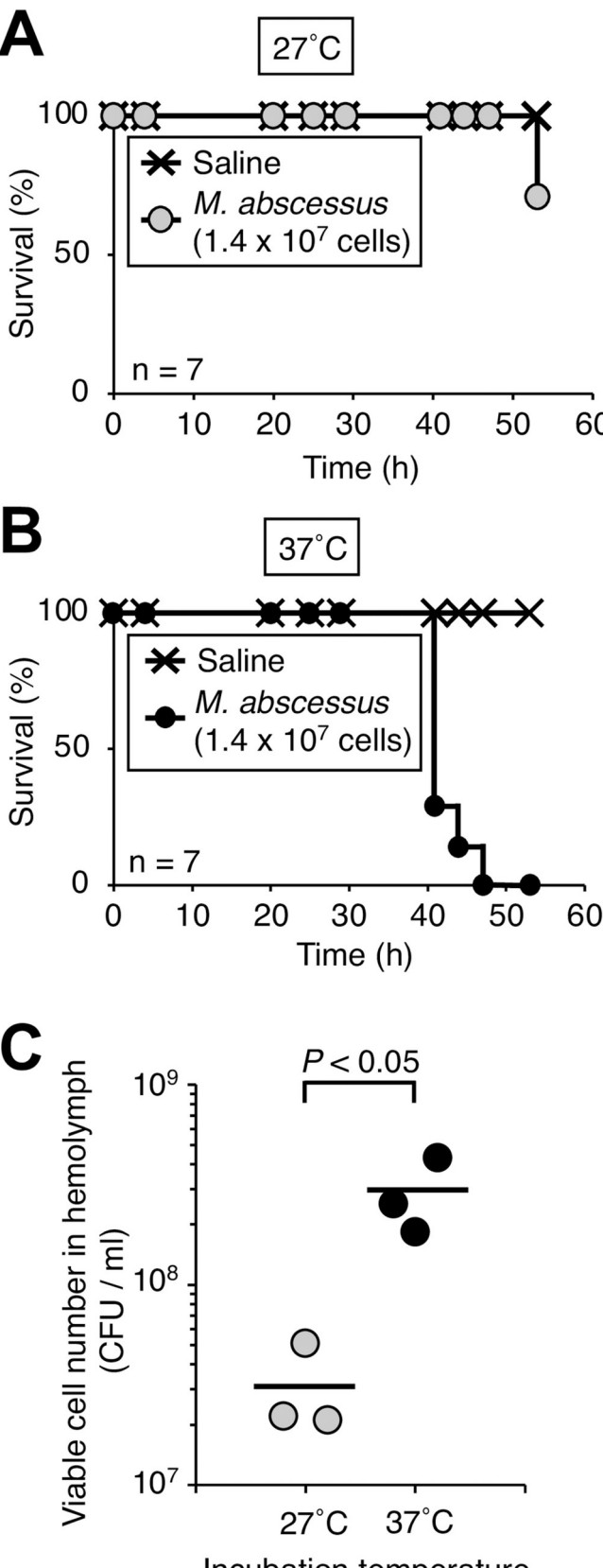

**Fig 1. Effects of temperature on the virulence of *M. abscessus* subsp. *abscessus* ATCC19977 in silkworms.** Silkworms were injected with saline (50 μl) or *M. abscessus* subsp. *abscessus* ATCC19977 cell suspension ($1.4 \times 10^7$ cells per 50 μl) and incubated at (**A**) 27°C and (**B**) 37°C. The curves were drawn by the Kaplan-Meier method. Seven silkworms were used per group. (**C**) Silkworms were injected with *M. abscessus* subsp. *abscessus* ATCC19977 cell suspension ($2.9 \times 10^8$ cells per 50 μl) and incubated at 27°C and 37°C. Silkworm hemolymph was harvested at 18 h post-infection. Statistically significant differences between groups were evaluated using the Student *t*-test. Three silkworms were used per group.

## Evaluating the virulence of *M. abscessus* subsp. *abscessus* clinical isolates against silkworms

We next determined the $LD_{50}$ values of *M. abscessus* subsp. *abscessus* clinical isolates using the silkworm infection model to compare their virulence. Seven clinical isolates were obtained from sputum samples of patients infected with *M. abscessus* subsp. *abscessus*. Their $LD_{50}$ values ranged from $3.1 \times 10^6$ to $2.9 \times 10^7$ cells per larva, with the $LD_{50}$ value of the Mb-17 isolate being the lowest (Fig 5). The $LD_{50}$ value of the Mb-17 isolate was 9-fold lower than that of the Mb-10 isolate. The viable cell count of the Mb-17 isolate in the silkworm hemolymph at 18 h post-infection was higher than that of the Mb-10 isolate (Fig 6). These results suggest that the Mb-17 isolate has higher virulence against silkworms than the Mb-10 isolate.

## Evaluating the cytotoxicities of *M. abscessus* subsp. *abscessus* clinical isolates against human THP-1–derived macrophages

Human THP-1-derived macrophages can adhere to the polystyrene surface of a 96-well plate. Chemical induced cell detachment from the well correlates with cell death [29]. We next determined the cytotoxicities of the Mb-10 and Mb-17 isolates against THP-1–derived macrophages by monitoring cell detachment. The number of macrophages attached to the well was decreased by infection with these strains at a multiplicity of infection of 50 (Fig 7). The Mb-17

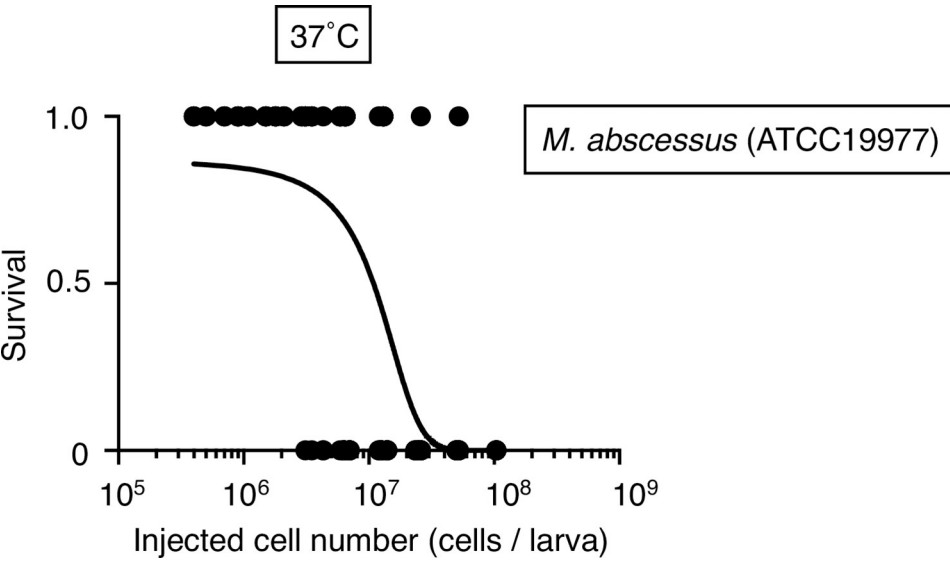

**Fig 2. Determination of the *M. abscessus* subsp. *abscessus* ATCC19977 $LD_{50}$ in silkworms.** Silkworms were injected with saline (50 μl) or *M. abscessus* subsp. *abscessus* ATCC19977 cell suspension ($4 \times 10^5$–$1 \times 10^8$ cells per 50 μl) and incubated at 37°C for 2 days. The numbers of live and dead silkworms are indicated as 1 and 0, respectively. The curve represents data from 6 independent experiments combined in a simple logistic regression model. One hundred twenty-nine silkworms were used in the experiment.

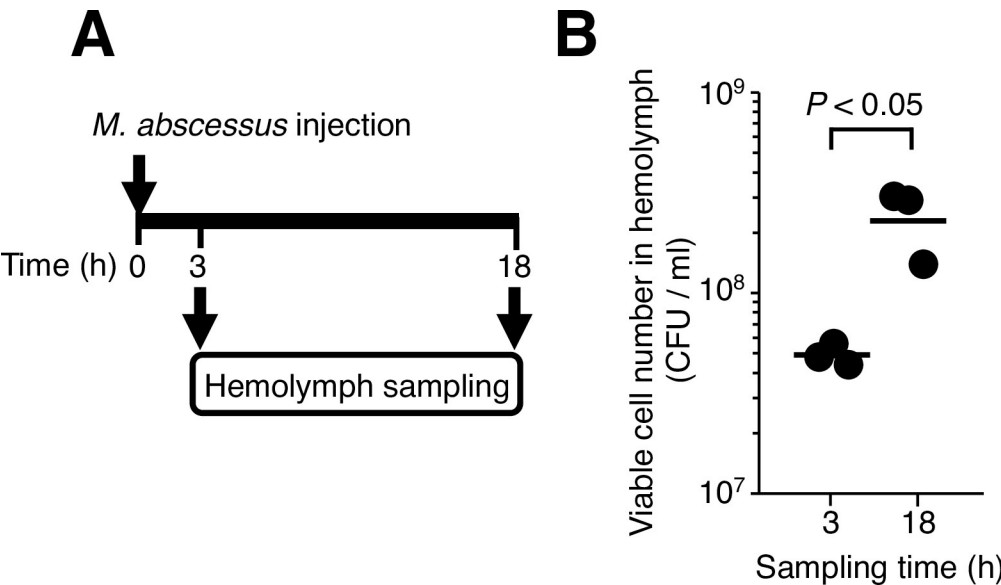

**Fig 3. Increase of *M. abscessus* subsp. *abscessus* ATCC19977 viable cell counts in silkworms.** (**A**) Experiment schematic. (**B**) Silkworms were injected with *M. abscessus* subsp. *abscessus* ATCC19977 cell suspensions ($5 \times 10^8$ cells per 50 µl) and incubated at 37˚C. Silkworm hemolymph was harvested at 3 or 18 h post-infection. The viable cell number of *M. abscessus* subsp. *abscessus* was measured by counting the colony-forming units (CFU). Statistically significant differences between groups were evaluated using the Student *t*-test. Three silkworms were used per group.

isolate led to a decrease in the number of attached macrophages compared with the Mb-10 isolate (Fig 7). The result suggests that the Mb-17 isolate, which was identified as a highly virulent strain using the silkworm infection model, induces a greater detachment of THP-1–derived macrophages during infection than the Mb-10 isolate.

## Discussion

In the present study, the virulence of *M. abscessus* subsp. *abscessus* clinical isolates was compared using a silkworm infection model. Among the 7 clinical isolates, the virulence, as determined by the $LD_{50}$, varied up to 9-fold. These results indicate that the *in vivo* silkworm evaluation system is useful for revealing the virulence of *M. abscessus* subsp. *abscessus* clinical isolates with a short time period (2 days).

*M. abscessus* subsp. *abscessus*-infected silkworms incubated at 37˚C were more sensitive to infection than those reared at 27˚C. We assumed that this difference was due to both high-temperature stress in silkworms and the optimal growth temperature for *M. abscessus* subsp. *abscessus*. Hosoda *et al.* reported establishing a silkworm infection model for evaluating anti-mycobacterial compounds [23]. Here, we demonstrated that *M. abscessus* subsp. *abscessus* grows in the silkworm hemolymph and compared the virulence of several clinical isolates. Our findings are important toward validating the usefulness of the silkworm infection model for estimating the virulence of *M. abscessus* subsp. *abscessus* clinical isolates. The $LT_{50}$ value, which is the incubation time required to kill half of the silkworms in a group, differed slightly among time-course experiments. Therefore, a secondary evaluation to determine the $LD_{50}$ values based on multiple dose-dependent experiments is needed.

*M. abscessus* subsp. *abscessus* virulence may correlate with severity infection [5, 9]. Therefore, understanding *M. abscessus* subsp. *abscessus* clinical isolate virulence is useful information for infection control. *M. abscessus* subsp. *abscessus* exhibited a different extent of

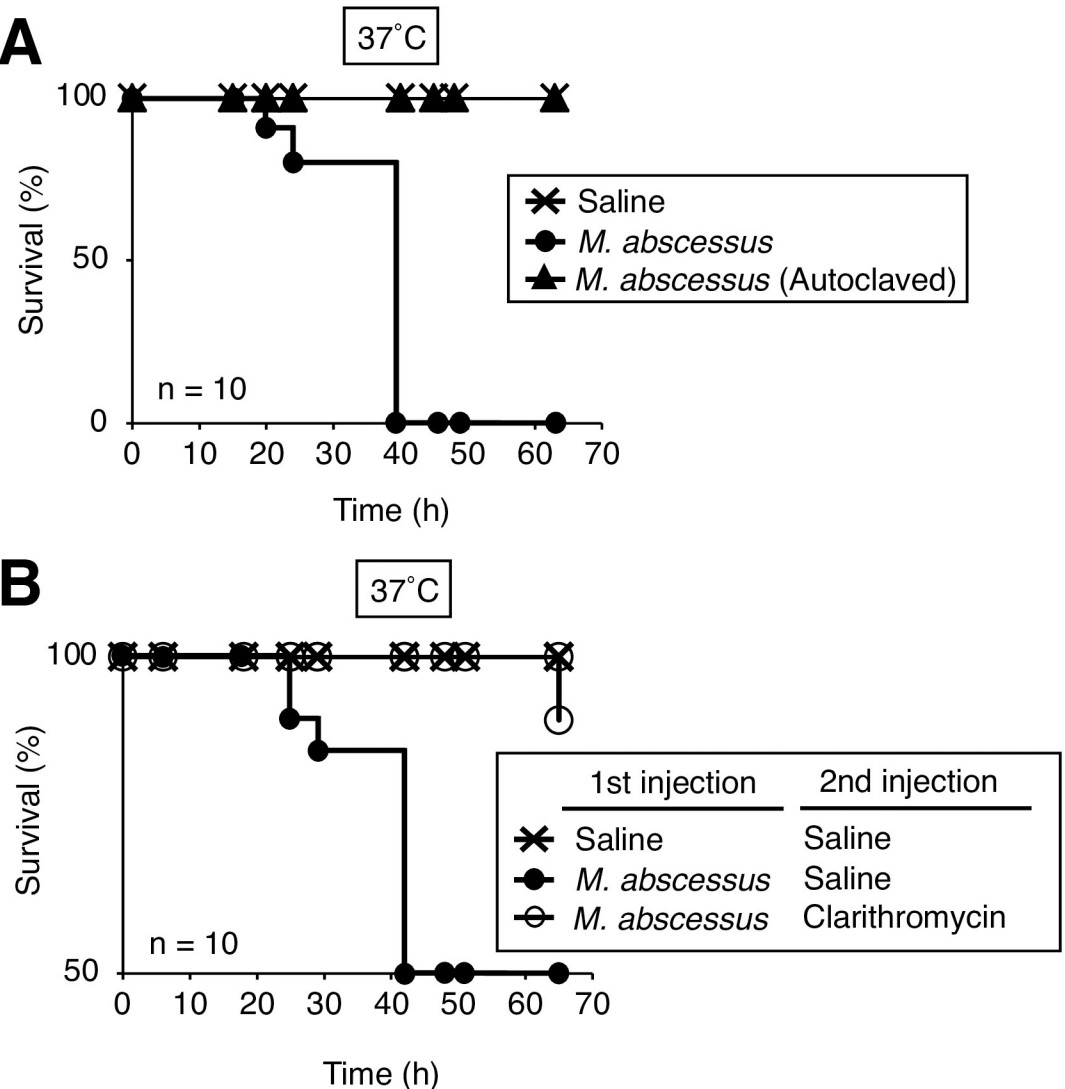

**Fig 4. Effects of autoclaved cells and antibacterial treatment in silkworms infected with *M. abscessus* subsp. *abscessus*.** (**A**) Silkworms were injected with either saline (50 μl), *M. abscessus* subsp. *abscessus* ATCC19977 cell suspension (1.1 x 10$^7$ cells per 50 μl), or autoclaved *M. abscessus* subsp. *abscessus* ATCC19977 cell suspension and incubated at 37°C. Ten silkworms were used per group. (**B**) Silkworms were injected with either saline (50 μl) or *M. abscessus* subsp. *abscessus* ATCC19977 cell suspension (6.3 x 10$^7$ cells per 50 μl) followed by clarithromycin (25 μg g$^{-1}$ larva). The number of surviving silkworms following incubation at 37°C was measured for 66 h. Statistically significant differences between groups were evaluated using a log-rank test based on the curves by the Kaplan-Meier method. Ten silkworms were used per group.

virulence among clinical isolates in the silkworm model. We demonstrated that the silkworm infection model with *M. abscessus* subsp. *abscessus* is advantageous for quantitative determination of clinical isolate virulence by calculating the LD$_{50}$ values within 2 days. Moreover, the LD$_{50}$ values among the clinical isolates differed up to 9-fold. The Mb-17 isolate was the most virulent against silkworms among the *M. abscessus* subsp. *abscessus* clinical isolates used in this study. We hypothesized that the Mb-17 isolates harbor virulence-related genes that enhance the infection process. Moreover, the cytotoxicity of the Mb-17 isolate against human THP-1–derived macrophages was higher than that of the Mb-10 isolate. These results suggest that the Mb-17 isolate, which is highly virulent in silkworm infection model, is also highly cytotoxic to human macrophages. Future studies will include experiments aimed at revealing

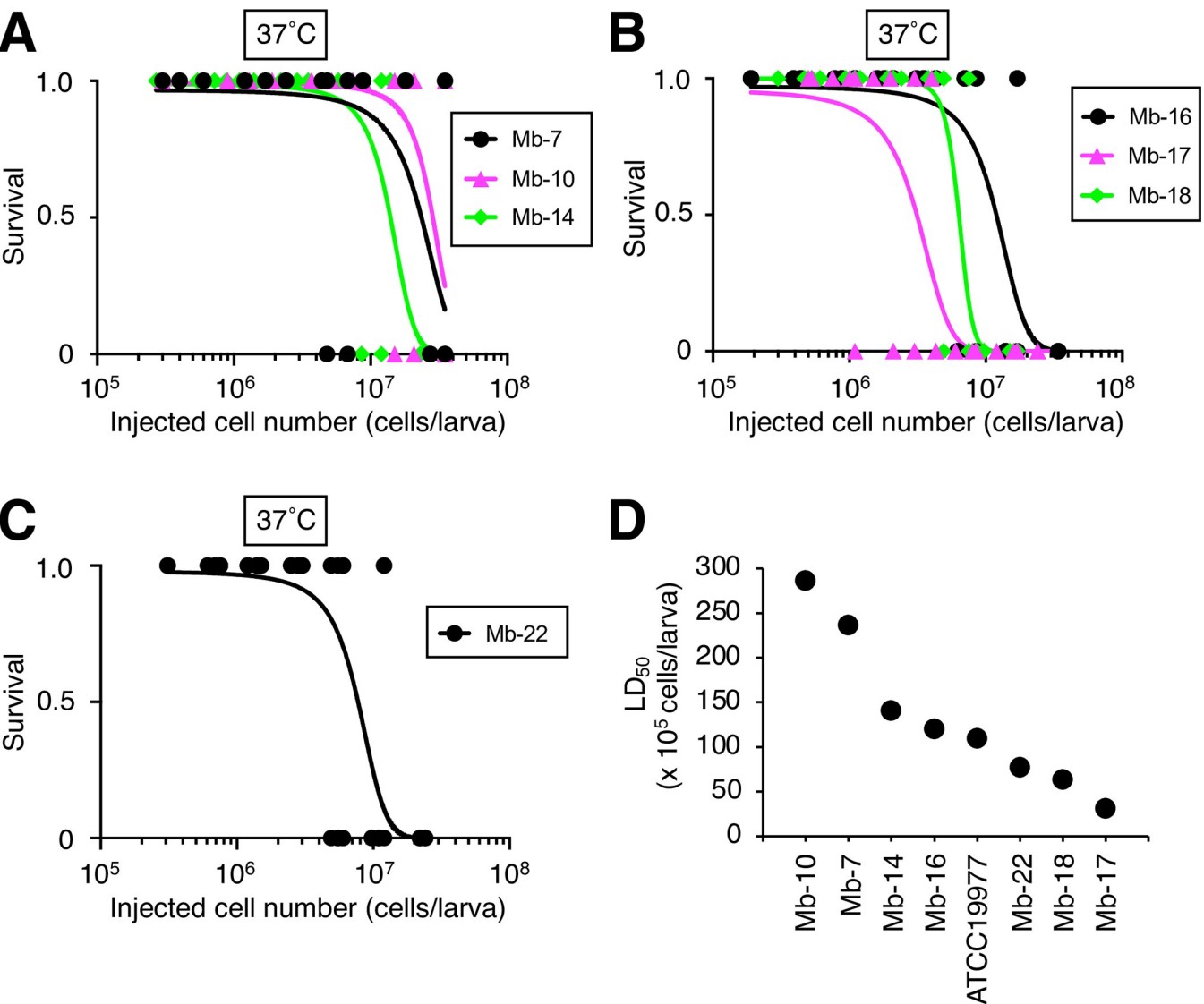

**Fig 5. Comparison of virulence among *M. abscessus* subsp. *abscessus* clinical isolates in a silkworm infection model.** (**A-C**) Silkworms were injected with either saline (50 μl), *M. abscessus* subsp. *abscessus*, Mb-7, Mb-10, Mb-14, Mb-16, Mb-17, Mb-18, or Mb-22 cell suspensions (2 x $10^5$–3.5 x $10^7$ cells per 50 μl) and incubated at 37˚C. Live and dead silkworms are indicated as 1 and 0, respectively. Curves represent data from 3 independent experiments combined in a simple logistic regression model. In each experiment, 39–54 silkworms were used. (**D**) Plot of $LD_{50}$ values determined from A-C.

the relationship between the information on disease parameters in patients and the virulence against silkworms. The virulence genes of several pathogens have been identified by mutant screening using silkworm infection models from a mutant library [19–22]. The method for constructing *M. abscessus* subsp. *abscessus* gene-deletion mutants is well established [32–34]. Further studies are needed to determine the virulence factors harbored by the *M. abscessus* subsp. *abscessus* Mb-17 isolate that are responsible for its virulence in silkworms.

*M. abscessus* vertebrate infection models using a zebrafish, *Danio rerio*, mice, and a tadpole, *Xenopus laevis*, have been reported [33, 35, 36]. These infection models are used to evaluate anti-mycobacterial drugs and virulence within 15 days. The rearing temperature may be related to the experimental period. For example, infected zebrafishes were reared at 28–32˚C. In this study, infected silkworms died under a rearing condition at 37˚C within 2 days but not

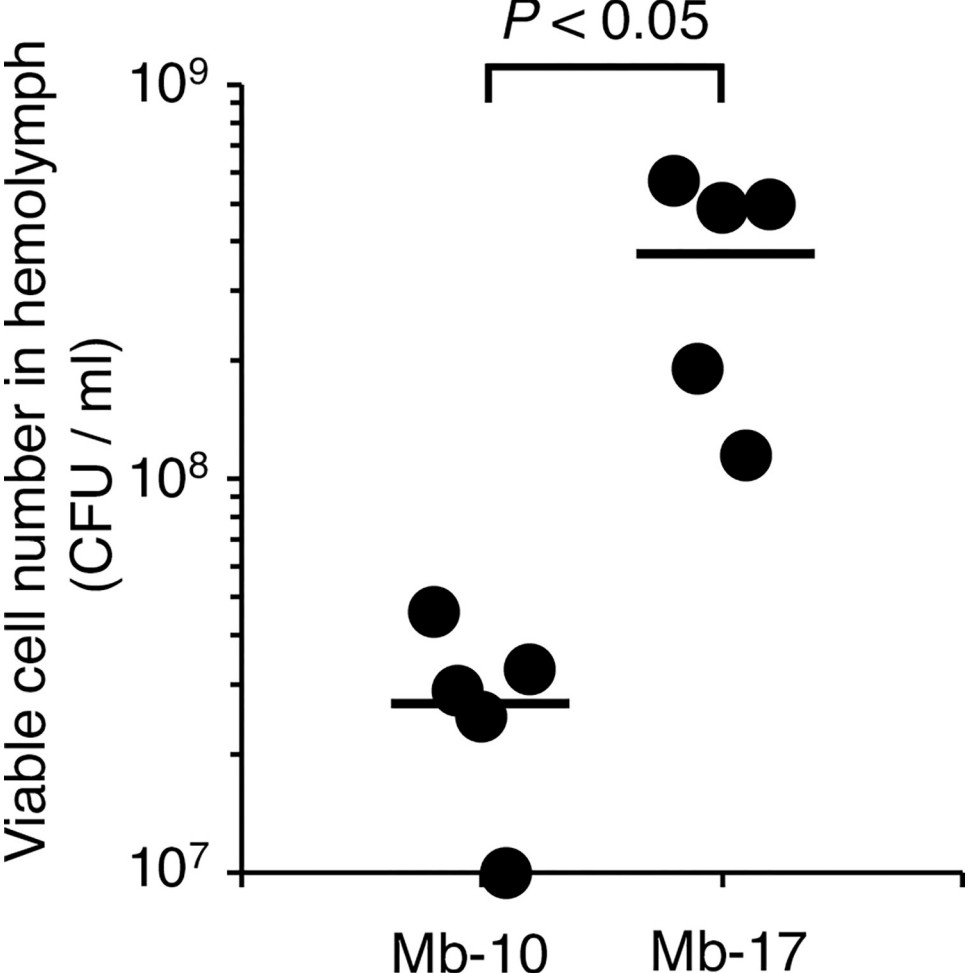

**Fig 6. Viable cell counts of *M. abscessus* subsp. *abscessus* Mb-10 and Mb-17 isolates in silkworms.** Silkworms were injected with saline (50 μl) or *M. abscessus* subsp. *abscessus* Mb-10 cell suspension (3.6 x $10^8$ cells per 50 μl) or Mb-17 cell suspension (3.8 x $10^8$ cells per 50 μl) and incubated at 37°C. Silkworm hemolymph was harvested at 18 h post-infection. Five silkworms were used per group. Statistically significant differences between groups were evaluated using the Student *t*-test.

at 27°C. Therefore, the silkworm has the benefit of being able to examine the infection experiments at 37°C. Because these model animals including a zebrafish are vertebrates, however, more ethical problems are associated with their use with respect to animal welfare. The 3Rs, replacement, refinement, and reduction, are important principles for experiments using mammals [16]. Silkworms are invertebrate animals with several advantages as an alternative model animal for infection experiments requiring a high number of animals. Moreover, *M. abscessus* subsp. *abscessus* virulence factors relevant to human pathogenicity could be identified using this silkworm infection model.

Our study has some limitations. First, the silkworm infection model established in this study is not a respiratory infection model because the bacterial cells were injected into silkworm hemolymph. Therefore, the silkworm infection model deviates significantly from *M. abscessus* infection in humans. Second, it is unknown how the bacteria kill silkworms. Therefore, silkworm tissues targeted by the bacterial cells should be determined by histopathological studies.

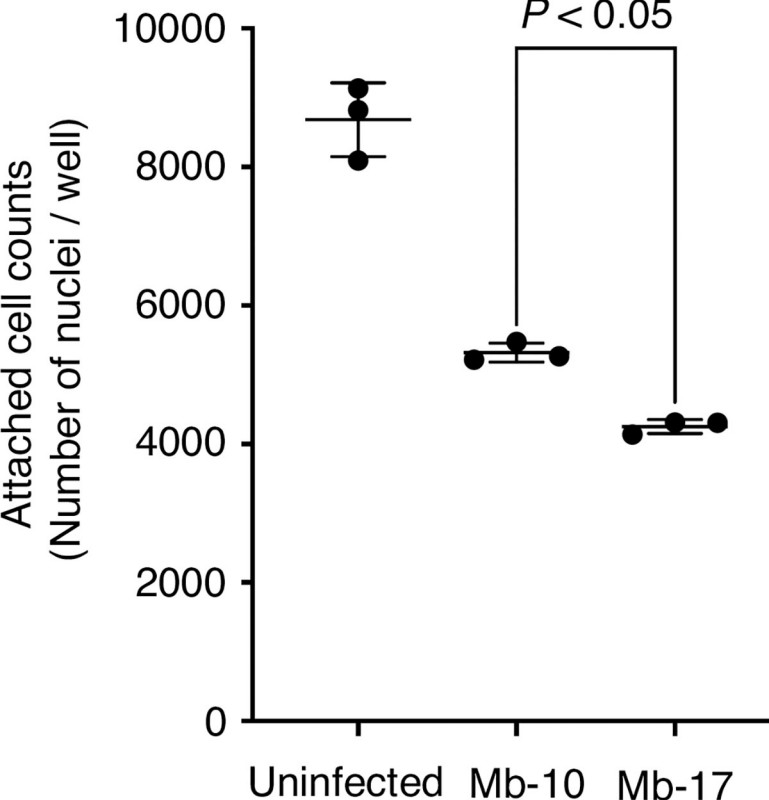

**Fig 7. Attached-cell counts of human THP-1 macrophages after infection with *M. abscessus subsp. abscessus* Mb-10 and Mb-17 isolates.** Attached-cell counts of human THP-1 macrophages at 48 h after infection with *M. abscessus* subsp. *abscessus* Mb-10 or Mb-17 cells at a multiplicity of infection of 50. The number of nuclei of macrophages attached to the well was calculated using High-Content Imaging System Operetta CLS with Harmony software. Three independent samples were used per group. Statistically significant differences between groups were evaluated using the Turkey test with one-way ANOVA.

## Conclusion

We propose that the silkworm infection model with *M. abscessus* subsp. *abscessus* is an advantageous assay system for determining the virulence of *M. abscessus* subsp. *abscessus* clinical strains in a short time period. The silkworm infection model may contribute to revealing the molecular mechanisms of *M. abscessus* subsp. *abscessus* infections.

## Supporting information

**S1 Dataset. Datasets included in this study.**
(XLSX)

## Acknowledgments

We thank Tae Nagamachi, Asami Yoshikawa, Yu Sugiyama, Eri Sato, and Asuka Toshima (Meiji Pharmaceutical University) for their technical assistance rearing the silkworms. We also thank Maki Okuda, Sayaka Kashiwagi, and Ginko Kaneda for their assistance.

## Author Contributions

**Conceptualization:** Yasuhiko Matsumoto, Yoshihiko Hoshino.

**Data curation:** Yasuhiko Matsumoto, Hanako Fukano.

**Formal analysis:** Yasuhiko Matsumoto, Hanako Fukano.

**Funding acquisition:** Yasuhiko Matsumoto, Hanako Fukano, Yoshihiko Hoshino.

**Investigation:** Yasuhiko Matsumoto, Hanako Fukano.

**Methodology:** Hanako Fukano, Naoki Hasegawa, Yoshihiko Hoshino.

**Project administration:** Yasuhiko Matsumoto.

**Resources:** Naoki Hasegawa, Yoshihiko Hoshino.

**Supervision:** Yasuhiko Matsumoto, Yoshihiko Hoshino, Takashi Sugita.

**Writing – original draft:** Yasuhiko Matsumoto.

**Writing – review & editing:** Yasuhiko Matsumoto, Hanako Fukano, Naoki Hasegawa, Yoshihiko Hoshino, Takashi Sugita.

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
