## [Decision Letter · Decision Letter 0]

2 Nov 2022

PONE-D-22-23137Quantitative evaluation of Mycobacterium abscessus clinical isolate virulence using a silkworm infection modelPLOS ONE

Dear Dr. Matsumoto, 

Thank you for submitting your manuscript to PLOS ONE. After careful consideration, we feel that it has merit but does not fully meet PLOS ONE’s publication criteria as it currently stands. Therefore, we invite you to submit a revised version of the manuscript that addresses the points raised during the review process.

I concur with the assessment of both reviewers that your study is interesting and important. Please address all of the comments from both reviewers and revise the manuscript accordingly.

Please submit your revised manuscript by Dec 17 2022 11:59PM. If you will need more time than this to complete your revisions, please reply to this message or contact the journal office at plosone@plos.org. Please include the following items when submitting your revised manuscript:A rebuttal letter that responds to each point raised by the academic editor and reviewer(s). You should upload this letter as a separate file labeled 'Response to Reviewers'.A marked-up copy of your manuscript that highlights changes made to the original version. You should upload this as a separate file labeled 'Revised Manuscript with Track Changes'.An unmarked version of your revised paper without tracked changes. You should upload this as a separate file labeled 'Manuscript'.If applicable, we recommend that you deposit your laboratory protocols in protocols.io to enhance the reproducibility of your results. Protocols.io assigns your protocol its own identifier (DOI) so that it can be cited independently in the future. For instructions see: https://journals.plos.org/plosone/s/submission-guidelines#loc-laboratory-protocols. Additionally, PLOS ONE offers an option for publishing peer-reviewed Lab Protocol articles, which describe protocols hosted on protocols.io. Read more information on sharing protocols at https://plos.org/protocols?utm_medium=editorial-email&utm_source=authorletters&utm_campaign=protocols.

We look forward to receiving your revised manuscript.

Kind regards,

Thomas Byrd

Academic Editor

PLOS ONE

Journal Requirements:

“This study was 328 supported in part by grants from the Japan Agency for Medical Research and Development/Japan International Cooperation Agency (AMED) to YH (JP20fk0108064, JP20fk0108075, JP21fk0108093, JP21fk0108129, JP21fk0108608, JP21jm0510004, JP21wm0125007, JP21wm0225004, JP21wm0325003, JP22gm1610003, JP22wm0225022, JP22wm0325054), to HF (JP22wm0325054), and Y.M. (JP22wm0325054); and for Scientific Research (C) to YM (JP20K07022) from the Japan Society for the Promotion of Science (JSPS).The funders had no role in the study design, data collection, data analysis, decision to publish, or preparation of the manuscript.”

“This study was supported in part by grants from the Japan Agency for Medical Research and Development/Japan International Cooperation Agency (AMED) to YH (JP20fk0108064, JP20fk0108075, JP21fk0108093, JP21fk0108129, JP21fk0108608, JP21jm0510004, JP21wm0125007, JP21wm0225004, JP21wm0325003, JP22gm1610003, JP22wm0225022, JP22wm0325054), to HF (JP22wm0325054), and Y.M. (JP22wm0325054); and for Scientific Research (C) to YM (JP20K07022) from the Japan Society for the Promotion of Science (JSPS). The funders had no role in the study design, data collection, data analysis, decision to publish, or preparation of the manuscript.”

Reviewers' comments:

Reviewer's Responses to Questions

**Comments to the Author**

1. Is the manuscript technically sound, and do the data support the conclusions?

Reviewer #1: Yes

Reviewer #2: Yes

2. Has the statistical analysis been performed appropriately and rigorously? 

Reviewer #1: No

Reviewer #2: Yes

3. Have the authors made all data underlying the findings in their manuscript fully available?

Reviewer #1: Yes

Reviewer #2: Yes

4. Is the manuscript presented in an intelligible fashion and written in standard English?

Reviewer #1: Yes

Reviewer #2: Yes

5. Review Comments to the Author

Reviewer #1: A nice study, easy to follow. I have only two text addition requests for the discussion and one statistical recalculation:

How accurate is hemolymph for assessing MABC burden? Presumably these bacteria can invade the tissue like they do in other model systems so there might be unaccounted burden missed by a more thorough method such as homogenisation. Please elaborate discussion to speculate if tissue invasion is expected and if the silk worm in amenable to histological study.

Line 308: zebrafish models can evaluate virulence < 7 days (Kremer lab) in embryos, or the 14-15 day time period is from an Oehlers lab paper using adult zebrafish. It is worth commenting around this paragraph on the temperature differences, the silk worm allows the study of virulence at 37 while zebrafish studies are in the 28-32 degree range.

Figure 7: please replace T test with ANOVA

Reviewer #2: Virulence of seven clinical isolates of Mycobacterium abscessus subspecies abscessus was investigated using silkworm infection model. Additional, using a standard/validated protocol, virulence of these isolates was also studied in human macrophage cell line THP-1. Overall, the studies were straightforward and systematically performed. Data were interpreted in a rigorous manner and conclusions are justified by the results. Below I have included recommendations for revisions to improve on clarity and accuracy. Also, I have recommended to include additional discussion on the limitations of these M. abscessus infection model.

MINOR COMMENTS:

1. Abstract: It is redundant to use M. abscessus subspecies M. abscessus throughout the abstract as only supspecies abscessus is used in the study. So, please delete ‘subspecies abscessus’ from the following lines (Line 25, another one in line 25, line 26, 27, 29).

2. Line 28: add ‘in humans’ after ‘infection’.

3. Line 30: replace ‘were determined using’ with ‘in’.

4. Line 33: delete ‘with’.

5. Line 46: replace ‘effects’ with ‘efficacies’.

6. Line 46-48: It is stated that ‘mice infected with MABC dies over several months’. This is completely untrue. Even the cited references show that it requires immunesuppression or immunecompromised mice to maintain MABC infection. So this sentence is factually incorrect and should be deleted. Replace it with the following sentence: ‘Existing mouse models of MABC infection require several weeks to complete a single study, which is not convenient especially for MABC virulence screening purposes, and thus the development of a model that permits a more rapid evaluation of MABC virulence is highly desirable.’

7. Animal models of a human disease have their strengths and limitations. A good and objective manuscript should include limitations of the model. The authors should include the fact that most of the Mycobacterium abscessus infections are acquired via aerosol and this model does not take that into account. Most of the Mycobacterium abscessus disease is pulmonary and this model cannot recapitulate that aspect. The formation of lesion in the lungs and immune response are distinct from that in the silkworm. Therefore, it is important to include limitations of the silkworm model. Then describe the unique strengths of this model, which is the rapid time in which virulence of M. abscessus can be studied. Also, a large number of silkworms can be used, which is important for screening studies.

8. NOTE that the study assesses ‘virulence’ and not ‘pathogenecity’ of M. abscessus isolates.

9. Figure 1: what is n? n should be described as the number of independent M. abscessus isolates so that it is not confused with the number of silkworms used. Also, the number of silkworms used should be included in the figure legend. Pathogenecity study should include progression of disease in terms of tissue pathology and disease phenotypes in the host as the disease evolves. Therefore, ‘virulence’ is the correct terms and ‘pathogenecity’ should not be used. Line 31: replace ‘pathogenic’ with ‘virulent’. Line 224: replace ‘pathogenecity’ with ‘virulence’. Line 233: replace ‘pathogenecity’ with ‘virulence’. Line 295: replace ‘pathogenic’ with ‘virulent’.

10. Figure 2. In the figure legend ‘n’ should be defined as the number of silkworms used.

11. Figure 3. What is ‘n’? Is it the number of silkworms used or M. abscessus isolates?

12. Figure 4: Please define ‘n’. It represents the number of what?

13. Figure 5: Please define ‘n’.

14. Figure 6: Please define ‘n’.

15. Line 97: insert ‘were’ after ‘silkworms’.

16. Line 98: insert ‘ , and’ after ‘larva)’.

17. Line 99: Briefly describe how silkworms were injected with M. abscessus. What type of instrument/device was used to deliver M. abscessus?

18. Line 200: Insert ‘in humans’ after ‘infections’.

19. Line 288: replace ‘severe’ with ‘severity’.

20. Line 299: replace ‘was identified as a highly pathogenic strain using the’ with ‘is highly virulent in’

21. Line 303: delete ‘avirulent’. This term is unnecessary here.

22. Line 305: The cited reference #33 by Foreman et al is not appropriate here as random transposon mutagenesis is described in the publication rather than targeted gene deletion in MABC. This reference should be removed. More relevant references are [Nessar R, Reyrat JM, Davidson LB, Byrd TF. Deletion of the mmpL4b gene in the Mycobacterium abscessus glycopeptidolipid biosynthetic pathway results in loss of surface colonization capability, but enhanced ability to replicate in human macrophages and stimulate their innate immune response. Microbiology (Reading). 2011 Apr;157(Pt 4):1187-1195. doi: 10.1099/mic.0.046557-0. Epub 2011 Feb 3. PMID: 21292749.] and [Galanis C, Maggioncalda EC, Kumar P, Lamichhane G. Glby, Encoded by MAB_3167c, Is Required for In Vivo Growth of Mycobacteroides abscessus and Exhibits Mild β-Lactamase Activity. J Bacteriol. 2022 May 17;204(5):e0004622. doi: 10.1128/jb.00046-22. Epub 2022 Apr 5. PMID: 35380462; PMCID: PMC9112878.].

23. Line 308: insert ‘mice’ after ‘Danio rerio’. Mice should be included in this list, otherwise this sentence will be factually incorrect.

6. PLOS authors have the option to publish the peer review history of their article (what does this mean?). If published, this will include your full peer review and any attached files.

Reviewer #1: No

Reviewer #2: No

---

## [Author Response · Author response to Decision Letter 0]

16 Nov 2022

Reviewer #1: A nice study, easy to follow. I have only two text addition requests for the discussion and one statistical recalculation:

How accurate is hemolymph for assessing MABC burden? Presumably these bacteria can invade the tissue like they do in other model systems so there might be unaccounted burden missed by a more thorough method such as homogenisation. Please elaborate discussion to speculate if tissue invasion is expected and if the silk worm in amenable to histological study.

According to the reviewer’s comment, we added the sentences in the Discussion section of the revised manuscript (Page 15, lines 324-329).

[Page 15, lines 324-329] 

Our study has some limitations. First, the silkworm infection model established in this study is not a respiratory infection model because the bacterial cells were injected into silkworm hemolymph. Therefore, the silkworm infection model deviates significantly from M. abscessus infection in humans. Second, it is unknown the reason to cause silkworm death by growing the bacteria in which tissues of the silkworm. Therefore, histological studies need to determine silkworm tissue which is more abundant bacterial cells.

Line 308: zebrafish models can evaluate virulence < 7 days (Kremer lab) in embryos, or the 14-15 day time period is from an Oehlers lab paper using adult zebrafish. It is worth commenting around this paragraph on the temperature differences, the silk worm allows the study of virulence at 37 while zebrafish studies are in the 28-32 degree range.

According to the reviewer’s comment, we added the sentence that the rearing temperature of the animals affects the experimental period.

[Page 15, lines 313-316]

The rearing temperature may be related to the experimental period. For example, infected zebrafishes were reared at 28-32˚C. In this study, infected silkworms died under a rearing condition at 37˚C within 2 days but not at 27˚C. Therefore, the silkworm has the benefit of being able to examine the infection experiments at 37˚C. 

Figure 7: please replace T test with ANOVA

Following the reviewer’s comment, we performed the Turkey test with one-way ANOVA in Figure 7 (Page 13, line 273).

[Page 13, line 273]

using the Turkey test with one-way ANOVA

Reviewer #2: Virulence of seven clinical isolates of Mycobacterium abscessus subspecies abscessus was investigated using silkworm infection model. Additional, using a standard/validated protocol, virulence of these isolates was also studied in human macrophage cell line THP-1. Overall, the studies were straightforward and systematically performed. Data were interpreted in a rigorous manner and conclusions are justified by the results. Below I have included recommendations for revisions to improve on clarity and accuracy. Also, I have recommended to include additional discussion on the limitations of these M. abscessus infection model.

According to the reviewer’s comment, we added the limitations in the new paragraph of the Discussion section (Page 15, lines 324-329).

[Page 15, lines 324-329]

Our study has some limitations. First, the silkworm infection model established in this study is not a respiratory infection model because the bacterial cells were injected into silkworm hemolymph. Therefore, the silkworm infection model deviates significantly from M. abscessus infection in humans. Second, it is unknown the reason to cause silkworm death by growing the bacteria in which tissues of the silkworm. Therefore, histological studies need to determine silkworm tissue which is more abundant bacterial cells.

MINOR COMMENTS:

1. Abstract: It is redundant to use M. abscessus subspecies M. abscessus throughout the abstract as only supspecies abscessus is used in the study. So, please delete ‘subspecies abscessus’ from the following lines (Line 25, another one in line 25, line 26, 27, 29).

Following the reviewer’s comment, we delete the description “subspecies abscessus” in the Abstract section of the revised manuscript.

2. Line 28: add ‘in humans’ after ‘infection’.

Following the reviewer’s comment, we added the description in the revised manuscript (Page 2, line 26).

3. Line 30: replace ‘were determined using’ with ‘in’.

Following the reviewer’s comment, we replaced the description in the revised manuscript (Page 2, line 28).

4. Line 33: delete ‘with’.

Following the reviewer’s comment, we delete the description in the revised manuscript.

5. Line 46: replace ‘effects’ with ‘efficacies’.

Following the reviewer’s comment, we replaced the description in the revised manuscript (Page 3, line 44).

6. Line 46-48: It is stated that ‘mice infected with MABC dies over several months’. This is completely untrue. Even the cited references show that it requires immunesuppression or immunecompromised mice to maintain MABC infection. So this sentence is factually incorrect and should be deleted. Replace it with the following sentence: ‘Existing mouse models of MABC infection require several weeks to complete a single study, which is not convenient especially for MABC virulence screening purposes, and thus the development of a model that permits a more rapid evaluation of MABC virulence is highly desirable.’

Thank you for pointing this out and kind suggestion. According to the reviewer’s suggestion, we deleted the sentence and added the suggested sentence in the revised manuscript (Page 3, lines 44-47).

[Page 3, lines 44-47]

Existing mouse models of MABC infection require several weeks to complete a single study, which is not convenient, especially for MABC virulence screening purposes. Thus, developing a model that permits a more rapid evaluation of MABC virulence is highly desirable.

7. Animal models of a human disease have their strengths and limitations. A good and objective manuscript should include limitations of the model. The authors should include the fact that most of the Mycobacterium abscessus infections are acquired via aerosol and this model does not take that into account. Most of the Mycobacterium abscessus disease is pulmonary and this model cannot recapitulate that aspect. The formation of lesion in the lungs and immune response are distinct from that in the silkworm. Therefore, it is important to include limitations of the silkworm model. Then describe the unique strengths of this model, which is the rapid time in which virulence of M. abscessus can be studied. Also, a large number of silkworms can be used, which is important for screening studies.

Thank you for your kind suggestion. According to the reviewer’s suggestion. We added the sentence of the limitations in this study of the Discussion section in the revised manuscript (Page 15, lines 324-329). 

[Page 15, lines 324-329]

Our study has some limitations. First, the silkworm infection model established in this study is not a respiratory infection model because the bacterial cells were injected into silkworm hemolymph. Therefore, the silkworm infection model deviates significantly from M. abscessus infection in humans. Second, it is unknown the reason to cause silkworm death by growing the bacteria in which tissues of the silkworm. Therefore, histological studies need to determine silkworm tissue which is more abundant bacterial cells.

8. NOTE that the study assesses ‘virulence’ and not ‘pathogenecity’ of M. abscessus isolates.

Thank you very much for your suggestion. We revised the manuscript according to the reviewer’s comment (Page 11, line 227, 236).

9. Figure 1: what is n? n should be described as the number of independent M. abscessus isolates so that it is not confused with the number of silkworms used. Also, the number of silkworms used should be included in the figure legend. Pathogenecity study should include progression of disease in terms of tissue pathology and disease phenotypes in the host as the disease evolves. Therefore, ‘virulence’ is the correct terms and ‘pathogenecity’ should not be used. Line 31: replace ‘pathogenic’ with ‘virulent’. Line 224: replace ‘pathogenecity’ with ‘virulence’. Line 233: replace ‘pathogenecity’ with ‘virulence’. Line 295: replace ‘pathogenic’ with ‘virulent’.

According to the reviewer’s comment, we replaced the words in the revised manuscript. Moreover, we changed the Figure legends of the revised manuscript (Page 9, lines 180, 183-184, Page 11, line 227, 236). 

10. Figure 2. In the figure legend ‘n’ should be defined as the number of silkworms used.

Following the reviewer’s comment, we described the number of silkworms used (Page 9, lines 190-191).

11. Figure 3. What is ‘n’? Is it the number of silkworms used or M. abscessus isolates?

Following the reviewer’s comment, we described the number of silkworms used (Page 10, lines 212-213).

12. Figure 4: Please define ‘n’. It represents the number of what?

Following the reviewer’s comment, we described the number of silkworms used (Page 11, lines 219-220, 225).

13. Figure 5: Please define ‘n’.

Following the reviewer’s comment, we described the number of silkworms used (Page 12, line 244).

14. Figure 6: Please define ‘n’.

Following the reviewer’s comment, we described the number of silkworms used (Page 12, lines 251-252).

15. Line 97: insert ‘were’ after ‘silkworms’.

Following the reviewer’s comment, we added the word in the revised manuscript (Page 5, line 98).

16. Line 98: insert ‘ , and’ after ‘larva)’.

Following the reviewer’s comment, we added the word in the revised manuscript (Page 5, line 99).

17. Line 99: Briefly describe how silkworms were injected with M. abscessus. What type of instrument/device was used to deliver M. abscessus?

According to the reviewer’s comment, we added the sentence in the revised manuscript (Page 5, lines 96-98).

[Page 5, lines 96-98]

A 50-μl of sample solutions was administered to the silkworm hemolymph by injecting the silkworm dorsally using a 1-ml tuberculin syringe (Terumo Medical Corporation, Tokyo, Japan).

18. Line 200: Insert ‘in humans’ after ‘infections’.

Following the reviewer’s comment, we added the word in the revised manuscript (Page 10, line 202).

19. Line 288: replace ‘severe’ with ‘severity’.

Following the reviewer’s comment, we replaced the word in the revised manuscript (Page 14, line 292).

20. Line 299: replace ‘was identified as a highly pathogenic strain using the’ with ‘is highly virulent in’

Following the reviewer’s comment, we replaced the sentence in the revised manuscript (Page 14, line 303).

21. Line 303: delete ‘avirulent’. This term is unnecessary here.

Following the reviewer’s comment, we deleted the word in the revised manuscript.

22. Line 305: The cited reference #33 by Foreman et al is not appropriate here as random transposon mutagenesis is described in the publication rather than targeted gene deletion in MABC. This reference should be removed. More relevant references are [Nessar R, Reyrat JM, Davidson LB, Byrd TF. Deletion of the mmpL4b gene in the Mycobacterium abscessus glycopeptidolipid biosynthetic pathway results in loss of surface colonization capability, but enhanced ability to replicate in human macrophages and stimulate their innate immune response. Microbiology (Reading). 2011 Apr;157(Pt 4):1187-1195. doi: 10.1099/mic.0.046557-0. Epub 2011 Feb 3. PMID: 21292749.] and [Galanis C, Maggioncalda EC, Kumar P, Lamichhane G. Glby, Encoded by MAB_3167c, Is Required for In Vivo Growth of Mycobacteroides abscessus and Exhibits Mild β-Lactamase Activity. J Bacteriol. 2022 May 17;204(5):e0004622. doi: 10.1128/jb.00046-22. Epub 2022 Apr 5. PMID: 35380462; PMCID: PMC9112878.].

Thank you very much for your suggestion. According to the reviewer’s comment, we deleted the reference 33 in the previous manuscript and added the new references in the revised manuscript (Page 15, line 308, pages 21-22, lines 442-446, page 22, lines 451-454).

23. Line 308: insert ‘mice’ after ‘Danio rerio’. Mice should be included in this list, otherwise this sentence will be factually incorrect.

Following the reviewer’s comment, we added the word in the revised manuscript (Page 15, line 331).

---

## [Editor Report · Decision Letter 1]

23 Nov 2022

Quantitative evaluation of Mycobacterium abscessus clinical isolate virulence using a silkworm infection model

PONE-D-22-23137R1

Dear Dr. Matsumoto:

We’re pleased to inform you that your manuscript has been judged scientifically suitable for publication and will be formally accepted for publication once it meets all outstanding technical requirements.

Kind regards,

Thomas Byrd

Academic Editor

PLOS ONE
---

## [Editor Report · Acceptance letter]

12 Dec 2022

PONE-D-22-23137R1 

Quantitative evaluation of *Mycobacterium abscessus* clinical isolate virulence using a silkworm infection model 

Dear Dr. Matsumoto:

I'm pleased to inform you that your manuscript has been deemed suitable for publication in PLOS ONE. Congratulations! Your manuscript is now with our production department. 

Kind regards, 

on behalf of

Dr. Thomas Byrd 

Academic Editor

PLOS ONE